# Elastin-Dependent Aortic Heart Valve Leaflet Curvature Changes During Cyclic Flexure

**DOI:** 10.3390/bioengineering6020039

**Published:** 2019-05-07

**Authors:** Melake D. Tesfamariam, Asad M. Mirza, Daniel Chaparro, Ahmed Z. Ali, Rachel Montalvan, Ilyas Saytashev, Brittany A. Gonzalez, Amanda Barreto, Jessica Ramella-Roman, Joshua D. Hutcheson, Sharan Ramaswamy

**Affiliations:** Department of Biomedical Engineering, Florida International University, Miami, FL 33174, USA; mtesf003@fiu.edu (M.D.T.); amirz013@fiu.edu (A.M.M.); dchap015@fiu.edu (D.C.); aali056@fiu.edu (A.Z.A.); rmont065@fiu.edu (R.M.); isaytash@fiu.edu (I.S.); bgonz049@fiu.edu (B.A.G.); abarr247@fiu.edu (A.B.); jramella@fiu.edu (J.R.-R.); jhutches@fiu.edu (J.D.H.)

**Keywords:** aortic valve, calcification, elastin degradation, leaflet, curvature, biomarker, early detection

## Abstract

The progression of calcific aortic valve disease (CAVD) is characterized by extracellular matrix (ECM) remodeling, leading to structural abnormalities and improper valve function. The focus of the present study was to relate aortic valve leaflet axial curvature changes as a function of elastin degradation, which has been associated with CAVD. Circumferential rectangular strips (L × W = 10 × 2.5 mm) of normal and elastin-degraded (via enzymatic digestion) porcine AV leaflets were subjected to cyclic flexure (1 Hz). A significant increase in mean curvature (*p* < 0.05) was found in elastin-degraded leaflet specimens in comparison to un-degraded controls at both the semi-constrained (50% of maximum flexed state during specimen bending and straightening events) and fully-constrained (maximally-flexed) states. This significance did not occur in all three flexed configurations when measurements were performed using either minimum or maximum curvature. Moreover, the mean curvature increase in the elastin-degraded leaflets was most pronounced at the instance of maximum flexure, compared to un-degraded controls. We conclude that the mean axial curvature metric can detect distinct spatial changes in aortic valve ECM arising from the loss in bulk content and/or structure of elastin, particularly when there is a high degree of tissue bending. Therefore, the instance of maximum leaflet flexure during the cardiac cycle could be targeted for mean curvature measurements and serve as a potential biomarker for elastin degradation in early CAVD remodeling.

## 1. Introduction

Calcific aortic valve disease (CAVD) is characterized by pathological remodeling of the aortic valve leaflets, fibrotic tissue formation, and calcified mineral deposition [1]. CAVD is the most frequent condition that necessitates surgical valve replacement [2,3]. In developed nations, an alarming rate of incidence of CAVD is projected, from 2.5 million cases in 2000 to approximately 4.5 million in 2030 [4]. Worldwide significance is also projected to reach an epidemic level with an estimated third of the global population aged 65 exhibiting at least early clinical signs of CAVD [5]. Risk factors for CAVD include: Older age, genetics, use of tobacco products, rheumatic fever and high blood pressure [6]. Untreated CAVD leads to valve malfunction typically via narrowing or stenosis of the aortic valve, which substantially augments the workload of the left ventricle. This leads to heart failure with current estimates of about 17,000 resulting deaths/year in the United States [7]. Surgery or transcatheter artificial valve replacement procedures are the only viable intervention currently available. Many patients cannot undergo surgery either due to high mortality risks or adverse responses to required anticoagulants when a mechanical valve is employed. Bioprosthetic replacement valves made of animal tissues do not require blood thinners, but typically last only between 8 to 15 years [8]. More effective management of CAVD is needed, including the development of pharmaceutical interventions. Potential non-invasive therapeutic targets have been identified [1], but clinical utility of these treatment options will require early diagnosis prior to irreversible gross valvular remodeling.

Previous investigations have indicated that there may be a correlation between elastin degradation and calcification in several cardiovascular disorders [1,9,10]. For example, Hinton et al. [9] have shown that the onset of CAVD is associated with aortic valve extracellular matrix (ECM) remodeling events, specifically, elastin degradation [4]. The degradation is caused by elastolytic enzymes including matrix metallopeptidases and cathepsins in the pathological ECM remodeling of the valves either via induction of fibrosis or by accelerated degradation of elastin fibers [4]. Elastin, which is predominant on the ventricularis layer proximal to the left ventricle, enhances full aortic valve closure via stretching during the end-diastolic phase of the cardiac cycle during which time the leaflets are subjected to transmembrane pressure stresses. Elastin also permits sufficient bending and re-coil of the aortic valve leaflets during the systolic and early diastolic phases respectively. This ensures that minimal energy losses are expended for the heart to pump blood to the systemic circulation, and that instantaneously after, the leaflets have efficiently transitioned to the coaptation state. 

In the current study, we sought to determine if a change in aortic valve leaflet tissue structure due to elastin degradation could be detected and subsequently quantified by curvature measurements. Beyond standard echocardiography for routine diagnosis of valve diseases, when greater spatial and/or temporal resolution is needed, such as for example for interventional and surgical planning purposes, transoesophageal echocardiography or multi-detector computed tomography are used in the acquisition of 3-dimensional heart valve geometry, with each modality having its own merits. For example, transoesophageal echocardiography is able to provide detailed spatial information on the distribution of calcification on the aortic valve and its surrounding vasculature. On the other hand, multi-detector computed tomography can provide detection of highly-tortuous geometries, with high spatial resolution in the order of 500 µm [11] to capture 3-D native aortic valve geometries with high-fidelity. Collectively, while the selection of imaging modalities for a given patient will be specific to the end-objectives, quantification of leaflet curvature in a fairly straightforward manner does currently appear to be feasible. Aortic valve leaflet curvature can be easily measured from imaging modalities such as echocardiography or Computer-Aided Tomography (CT)-derived images. Therefore, detection of changes in curvature based-metrics of the aortic valve leaflet due to abnormal elastin remodeling or loss could serve as a potential biomarker for early detection of CAVD. 

## 2. Materials and Methods

### 2.1. Tissue Sample Preparation

Following the protocol by Butcher et al. [12] for porcine valvular tissue isolation, young porcine hearts (~5 months old) were harvested from a local abattoir (Mary’s Ranch, Miami, FL, USA) from which the aortic valves were resected (n = 3 valves). The valves, along with the aortic root, were placed in cold 1X phosphate buffered saline solution (PBS) and transported to the laboratory. From each aortic valve, the three leaflets were excised from the aortic root and individually placed in a protease inhibitor (PI) solution prepared by dissolving a PI tablet (Sigma-Aldrich, St. Louis, MO, USA) in 50 mL PBS, so as to decelerate naturally-occurring degradation activity. Tissue sectioning began with cutting rectangular strips (10 mm +/− 0.1 mm × 2.5 mm +/− 0.5 mm) from the belly region of the valve with the long axis oriented in the circumferential direction. The remaining surrounding tissue was thoroughly minced, mixed, and subsequently divided into 5 equal parts. Storage conditions for, all tissues were maintained at 4 °C while immersed in the protease inhibitor solution. 

#### Elastin Degradation

Each rectangular tissue strip that was subjected to cyclic flexure (Section 2.2) and curvature assessment (Section 2.2) subsequently underwent elastin degradation, that is, each sample served as its own control. Elastin tissue degradation was performed with the use of an elastase powder (Bio Basic Inc., Toronto, ON, Canada) dissociated in de-ionized (DI) water [13]. In brief, 0.5 mg/mL of elastase solution was prepared in DI water by gently shaking the solution to maintain dissociation of the powder. Elastase degradation of each strip and corresponding specimen of minced tissue was performed for 2 h at 37 °C with 5% CO_2_ and 95% humidity. A control group consisted of un-degraded leaflet tissue strips in PI solution, that is, zero hour or non-exposure to the aqueous elastase enzyme.

### 2.2. Cyclic Flexure Experiments

Aortic valve leaflet uniaxial flexure experiments and subsequent curvature measurements were conducted using a mechanical testing instrument available in our laboratory (Electroforce 2300 with Wintest 7 software control, TA instruments, New Castle, DE, USA). Just prior to each experiment, the tissues were washed with 1X PBS and immersed in cold protease inhibitor solution. The tissue samples were measured in length, width and thickness before testing. Each specimen was then carefully affixed between the two grips length-wise (Figure 1). Initially, a caliper was used to allow for a known set distance during the recorded footage, in this case the distance between the left and right side of the grips (Figure 1). The parameters used for the experiment were as follows: Frequency of 1 Hz, time of 30 seconds, valve effective length between the grips was 11 mm and bending distance was from 2.8 units to –5 units (negative displacement) for bending and from –5 units to 2.8 units for stretching (positive displacement).

A simple camera system (Model #: 8/A1864, Apple Inc., Cupertino, CA, USA) was mounted tangential to the edge of the tissue specimens to record the bending and straightening events during cyclic flexure. The camera was positioned to precisely detect the specimen bending process along the edge of the sample (i.e., a 1-dimensional (1-D) axial edge measurement). Ten cycles of pre-conditioning were first carried out before data collection. Subsequently, each specimen was video-recorded during an entire cyclic flexure event, that is, from the straight specimen configuration, to the initiation of bending until the two sample grips were 0.5 cm apart and subsequently until the specimens returned to its original straight position. The video tracking was conducted on recorded frames using an in-house script based on the controlling parameters defined above (MATLAB, Mathworks, Natick, MA, USA). The total number of frames corresponding to the total recording time were distributed over the time and position for curvature calculations. The camera was set to record at 240 frames per second (FPS) totaling approximately 7,200 frames per video.

#### Curvature Assessment

Image frames collected from the cyclic flexure experiments were imported into an in-house script (MATLAB) to calculate the localized mean/minimum (min) and maximum (max) curvature (k_mean_, k_min_, k_max_) by fitting a line along the coordinates of the edges of the sample (Figure 2). For each frame the coordinates of the points along the edges were fitted to n^th^ degree polynomial as per the equation given below:(1)$$f\left(x\right)=a_{0}x^{n}+a_{1}x^{n-1}+\ldots +a_{n}$$

The curvature, k of the sample at each time point was then computed from the curvature equation [14]:(2)$$k=\frac{f\prime \prime }{{\left(1+{\left(f\prime \right)}^{2}\right)}^{3/2}}$$
where Y′ and Y″ refers to the first and second order derivatives of the polynomial equation, respectively. Since the curvature can vary widely along the edge of the sample the polynomial was fitted piecewise along the edge, resulting in a curvature value for every point (Figure 2C(i,ii,iii)). The minimum, maximum, and mean curvatures were then determined for each frame (Figure 2C(iv)). Validation of the computed curvature was previously conducted by comparing the in-house curvature code with three different circles of known radii. In all three cases, the computed curvature was found to be within 5% of the actual curvature of the circles.

The k_mean_, k_min_ and k_max_ metrics were computed at 5 temporal point during the flexure cycle. Specifically, these temporal points were as follows: Unconstrained or fully straight specimen positon prior to the bending event (UNC_b_), 50% constrained during bending or 50% of the maximum specimen flexure state during bending (50%-C_b_), the fully constrained or maximum specimen flexure state possible (FC), 50% constrained during straightening or 50% of the fully straightened specimen state during the straightening event (50%-C_s_) and finally, the fully straight or unconstrained specimen at the instance of complete straightening (UNC_s_). A total of 30 cycles of cyclic flexure were performed for each tissue specimen. The first 10 were excluded for pre-conditioning purposes. From the remaining 20 cycles, 5 were randomly chosen and their respective curvature was calculated at each of the bending phases and then averaged together. All 5 cycles were within 5% of each other for the minimum, maximum, and mean curvatures.

### 2.3. Elastin Assay

A commercially available Elastin assay kit (Fastin kit, Accurate Chemical Scientific, Westbury, NY, USA) was used to extract, solubilize and quantify elastin present in aortic valve leaflet tissues following the manufacturer’s protocol. The extraction of insoluble elastin from tissue samples was done by hot oxalis acid digestion; because tissue samples were very small, one extraction was necessary to digest all the sample present. Absorbance measurements were performed utilizing a spectrophotometer (Synergy HTX Multi-Mode reader, BioTek Instruments, Inc., Winooski, VT, USA). The elastin concentration was calculated for each of the un-degraded and elastin-degraded groups, using a calibration curve to convert measured absorbance values into actual concentrations. Finally, the elastin content of each sample was normalized with respect to its corresponding wet weight.

### 2.4. Elastin Structure

Unstained two-photon imaging using a home-built laser scanning microscopy system with broadband femtosecond excitation laser (Element 600, Femtolasers, Austria) was performed to evaluate elastin structure of porcine aortic valve leaflets (n = 3 leaflets), both before and after being subjected to a two-hour elastase degradation. The protocol for leaflet elastin degradation was identical to that utilized for degrading elastin in the tissue strips prior to cyclic flexure testing (Section 2.1). Microscopy was first carried out by scanning a laser beam with a pair of galvanometer mirrors (Thorlabs, Newton, NJ, USA) and directing it into a 20×/1 NA water immersion objective (Olympus, Japan), via a dichroic mirror (655spxr, Chroma, Bellows Falls, VT, USA) to separate two-photon signals from a reflected fundamental wavelength. Two photomultiplying tube photodetectors (Hamamatsu, Japan) with respective bandpass filters (400 nm central wavelength/40 nm bandwidth and 480 nm central wavelength/40 nm bandwidth) simultaneously recorded second harmonic generation and two-photon excitation fluorescence (TPEF) signals. Scanning and acquisition was controlled using a custom-written software (MATLAB) and a data acquisition board (National Instruments, Austin, TX, USA). Each image consisted of an average of ten 1000 × 1000-pixel frames acquired at 0.5 frames per second. The second harmonic generation (SHG)-signal was derived from the collagen fibers [15] in the leaflet tissues while TPEF-derived images consisted of signal intensities from elastin protein and cellular autofluorescence [13,16]. Images of un-degraded leaflets with a rich TPEF-signal but which were characterized by an absence of a fibrous structures were interpreted to be mainly autofluorescence and subsequently discarded. On the other hand, images that contained a fibrous network with TPEF-rich signal intensities were regarded as being elastin fibers and were thereby saved for comparison with images derived from the same leaflets after elastin degradation, at the corresponding spatial location. Prior to final evaluation of elastin structure, minor image processing was performed (ImageJ, NIH, Bethesda, MD, USA). Specifically, a 2-pixel mean radius filter was applied to each image, with subsequent square root of the intensity taken, and finally, brightness and contrast adjusted for the final visualization.

### 2.5. Histological Staining

The non-coronary leaflets from two porcine aortic heart valves were isolated. One leaflet was immersed in 0.5 mg/ml elastase solution for 2 h while the other leaflet was immediately fixed in 10% formalin. After two hours of elastase degradation, the degraded non-coronary leaflet was also immersed in 10% formalin. The leaflets were then rinsed with PBS and embedded in optimal cutting temperature (OCT). The leaflets were subsequently stored in –80 °C overnight. The next day, the embedded samples were cut depth-wise to a thickness of 10 µm/slice and mounted on glass slides (TruBond 380, Newcomer Supply, Middleton, WI, USA), which were allowed to dry at room temperature. Finally, the tissues were stained (Russel–Movat Pentachrome; American MasterTech Scientific, Item #: KTRMP, Lodi, CA, USA) to identify elastin and collagen distributions with the tissues. Each sample was then imaged and processed (ImageJ, NIH-Image, Bethesda, MD, USA) with auto-adjustment of the brightness and contrast.

### 2.6. Statistical Analysis

Curvature values were presented as the mean ± standard error of the mean (SEM). Significance in curvature measurements was determined via t-tests (GraphPad Software, San Diego, CA, USA) comparing the un-degraded controls to samples exposed to 2 h of enzymatic-solution at each of the 5 temporal-bending states evaluated, that is, UNC_b_, 50%-C_b_, FC, 50%C_s_ and UNC_s_. Sample size (n) for each of the 5 time-points/group was as follows: n = 9 leaflet strips/time-point for the un-degraded, control group and n = 6 leaflet strips/time-point for group that was subjected to the 2-hour elastase enzymatic degradation. A *p*-value of less than 0.05 (*p* < 0.05) between the two groups was considered to be statistically significant.

## 3. Results

### 3.1. Leaflet Shape Changes

Instantaneous image capture showed the changes in curvature at five levels (or temporal positions) of bending and straightening (Table 1). Visible differences between corresponding time-points were observed between the 2 h elastin-degraded and un-degraded groups. 

#### Curvature Computation

Mean, min, and max curvature for control and elastin-degraded (2 h of elastase degradation) porcine aortic valve leaflet specimens (between 6 and 9 strips/group) were computed at five different temporal positions during the cyclic flexure event (Table 2, Table 3 and Table 4). Mean curvature (k_mean_) between the two groups was found to be significantly higher for the elastin degraded tissue (*p* < 0.05) at the 50%-C_b_, FC and 50%-C_s_ flexure states. In the case of the k_min_ metric, only the 50%-C_b_ exhibited statistical significance between the groups, again with higher curvature in the degraded group. The magnitude of k_min_ values were relatively small compared to k_mean_ and k_max_. Finally, no significance (*p* > 0.05) between the groups was observed for k_max_. 

### 3.2. Leaflet Elastin Loss

The normalized elastin content (with respect to specimen wet weight) indicated that the elastin concentration in the elastin-degraded leaflet samples after 2 h of elastase exposure was 2.3-fold lower than the un-degraded samples (n = 3 specimens/degradation-group; Table 5).

### 3.3. Elastin Curvature Comparison

Elastin concentration was plotted against k_mean_, k_min_ and k_max_ for each temporal position of bending phase (Figure 3A–C respectively) for the two groups assessed in this study (0 Hours of elastase exposure (un-degraded controls) and 2 h of elastase exposure (elastase-degraded leaflet samples). A general trend of increase in curvature between the groups was observed with more elastin-degraded leaflet samples. Interestingly, the change in k_mean_ and k_max_ as a function of elastin loss was most pronounced when the leaflet tissue strips were at the maximum flexed state, that is, FC (Figure 3A,B). 

### 3.4. Elastin Structure

Un-degraded porcine aortic valve leaflets exhibited characteristic elastin network structure, with elastin fibers running in the radial direction (Figure 4A,C). On the other hand, after the leaflets were degraded with elastase, elastin was still present in the tissues but with a considerable loss in content observed (Figure 4B,D). In addition, there was clear disruption to elastin structure. Specifically, bulk aggregation of elastin was evident and was found to be randomly interspersed in the tissues. In addition, there was a complete loss in elastin fiber orientation and network.

### 3.5. Elastin Distribution

Histological evidence (Figure 5) revealed that an abundance of collagen was still visible within the leaflet after elastase-degradation, even though structure was severely disrupted (Figure 4). On the other hand, the elastin component of the non-coronary leaflet’s ECM was substantially lost, to the extent that the ventricularis layer was no longer visible (Figure 5).

## 4. Discussion and Conclusions

Matrix components such elastin and collagen are altered during early onset of CAVD [4,15]. Specifically, previous investigations have also described how elastolytic enzymes such as MMPs and cathepsins are involved in ECM remodeling by enhancing degradation of elastic fibers [1]. In particular, elastin degradation in aortic heart valves can serve as an initiating event in CAVD [16]. In the current work, we investigated a simple and rapid assessment for changes in aortic valve leaflet tissue curvature as a function of tissue elastin content and structure. The motivation for identifying such a diagnostic arises from the fact that while drug targets have been identified for the treatment of CAVD [17], early detection of the disease is still sorely needed. Aiming to understand these changes, we developed a methodology to quantify leaflet curvature change as a function of elastin loss and simultaneously, to identify alterations in the elastin structure when degraded. Monitoring leaflet curvature could thereby serve as a biomarker for early diagnosis of CAVD. 

Axial curvature (k_mean_, k_min_ and k_max_) was found to generally be increased in aortic leaflet tissue strips with degraded elastin. However, the k_min_ metric had very low magnitudes, thereby causing insufficient resolution for it to facilitate accurate distinctions between un-degraded and elastin-degraded specimens (Figure 3). Between the other two curvature metrics investigated (k_mean_ and k_max_), only k_mean_ exhibited significantly higher curvature states (*p* < 0.05) at multiple flexure configurations when the tissue specimens were elastin-degraded (Table 2). Specifically, these states occurred at 50%-C_b_, FC and 50%-C_s_. This important finding suggests that elastin loss due to abnormal longitudinal remodeling in the aortic valve can be detected during sufficient levels of bending in the leaflet as its shape configuration changes during the cardiac cycle. From a clinical diagnostic standpoint, the instance of maximum aortic valve leaflet flexure that occurs during the cardiac cycle can be targeted for medical image acquisition via echocardiography and/or CT. The axial 1-D mean curvature can be easily computed and monitored for changes with aging. Rather than focusing on the entire tri-leaflet geometry, a specific leaflet of the aortic valve could additionally be targeted, such as the non-coronary cusp for example, which is prone to calcification [18,19,20]. Moreover, abnormal aortic valve elastin remodeling events leading to loss in leaflet elastin content is likely to be accompanied by a loss in the radially-oriented elastin fiber network (Figure 4). Therefore, it is important to note that an increase in the axial 1-D mean leaflet curvature measurement in elastin-degraded leaflets is due to some combination of losses in both bulk elastin content and its structure. 

The present findings suggest that the elastase degradation protocol is not specific to elastin-alone and not surprisingly, given the physical interconnectedness of leaflet elastin-to-collagen fibers, led to disruption of the collagen structure as well (Figure 4). However, subsequent histological observations (Figure 5) confirmed that elastase-degradation led to the complete loss of elastin distribution within the leaflet whereas collagen was still very much present within the leaflet ECM. This result provides us with confidence that the curvature changes that initiate from the loss of elastin in the aortic valve, while collagen still remains, can be detectable, particularly at high leaflet flexure states. 

In conclusion, we have demonstrated that axial mean curvature of flexed, rectangular, circumferentially-aligned aortic heart valve leaflet strips (k_mean_ at 50%-C_b_, FC and 50%-C_s_ configurations) increases significantly (*p* < 0.05) when enzymatically-degraded after 2 h of exposure to elastase. This finding suggests that elastin remodeling events that occur as a precursor to CAVD could be monitored via echocardiography and/or by CT as a potential biomarker for early disease detection. 

## 5. Limitations

The study presented here represents preliminary findings and as such there were several limitations to this work. As can be observed (Figure 4), in addition to elastin, collagen structure was also lost during degradation of tissues with elastase. This is not entirely surprising since elastin fibers are under pre-tensile stress and serve to impose an intrinsic compressive stress on collagen fibers [21]. As elastin degrades, one would expect the collagen orientation to be disrupted (Figure 4). In our case, as stated, some collagen loss was observed (though much less compared to elastin (Figure 5)) in addition to severe alterations to the remaining collagen structure. Moreover, verification tests (unpublished observations) confirmed that elastase solution consisting of 0.5 mg/ml of elastase dissolved in DI water caused thickening of the leaflet tissue specimens. The thickening was even more pronounced when DI water-alone was used in a 2-hour mock degradation of un-degraded control leaflet tissue strips (data not shown). This finding suggests that the elastase solution utilized would not be able to maintain the same level of tissue thickening with varying levels of elastase concentration. Therefore, this thickening effect, coupled with the reported alterations on collagen content and structure, revealed that our investigation was limited by the lack of specificity to elastin degradation alone, which was not permissible via the 2-hour exposure of the leaflet tissues to 0.5 mg/ml aqueous elastase solution. Instead, three known properties of the leaflet tissues were modified after the enzymatic exposure: (i) Elastin (ii) collagen and (iii) the tissue thickness. Indeed, subsequent in vitro investigations will need to focus on adopting additional experimental controls that account for these three parameters, in order to establish a more conclusive interpretation that associates controlled loss of aortic valve elastin with its mean curvature. 

Another major limitation is that the curvature analyses herein focused on strips that were circumferentially dissected from the belly region of the porcine aortic leaflet which does not adequately represent valve leaflet geometry and associated in and out of plane deformations (2D and 3D curvature assessments) that occur during the cardiac cycle. Indeed, the commissures and more broadly the spatial locations where the leaflets are attached to the aorta are prone to high flexural states [22] and therefore serve as important valve anatomical targets for curvature computation. While these experiments are not straightforward, for in-plane curvature assessment, bi-axial cyclic flexure experiments would need to be considered to evaluate the resulting rectangular leaflet tissue strip curvature as a function of both the circumferential and radial directions. On the other hand, 3-dimensional curvature maps depicting the in vivo situation would still need to be further quantified via imaging the aortic valve leaflets while mounted in a flow loop that mimics the systemic hemodynamic conditions. These approaches represent important milestones that need to be considered as pre-cursors to eventual longitudinal in vivo curvature assessment of the aortic valve leaflets. In addition, valve tissue stiffness has been shown to be a function of the strain rate and frequency [23,24,25,26], specifically that a faster rate of deformation will yield a higher tissue stiffness. In the present study, leaflet strip curvature was assessed while the frequency was held constant at a physiologically-relevant magnitude of 1 Hz. It must be noted nonetheless that in the context of the in vivo situation, that changes in the heart rate could lead to different curvature responses in leaflets and hence, this effect has to be incorporated, for example, via a normalization process of curvature with respect to the heart rate, for more objective quantification of curvature. Furthermore, while we demonstrated that the flexed rather the straight configurations were more sensitive to differences between un-degraded and elastin-degraded specimens, without identifying specific phases in the cardiac cycle in which pronounced bending states occur, a direct application of the current findings to the clinic will not be possible. 

Another major limitation is that we only assessed elastin degradation after 2 h of elastase-exposure. The reason for this is that the utilized elastase solution (0.5 mg/mL) did not facilitate controlled and progressive loss of elastin in the tissues as a function of degradation times (data not shown). Therefore, a precise co-relation of k_mean_ with elastin content in the aortic valve ECM, over subtle, moderate and excessive loss of elastin (e.g., such as that previously described by Roach et al. [13]) would be necessary to determine a unique and precise correlation between k_mean_ and leaflet elastin loss.

Finally, although the relative changes in curvature assessment that was performed for each porcine tissue strip (all acquired from pigs at 5 months of age) before and after a severe tissue degradation protocol were observed, absolute differences in elastin content can vary considerably depending on the donor and even from the specific anatomical leaflet chosen (non-coronary, left coronary and right coronary cusps). This study did not attempt to sort tissue strips according to the donor source or anatomical cusp location. Note however, that proteolytic elastin degradation has been shown to initiate CAVD in mouse models [18], and fragmented elastin has been observed at sites of calcification in human aortic valve leaflets [1]. The degree of elastin fragmentation, however, has not been quantified over the course of CAVD progression. Temporal studies are difficult to perform in human patients due to the inability to biopsy aortic valve tissue. Hence, the current study provides important benchmarks of aortic valve leaflet curvature in tissues with intact and severely degraded elastin. Future studies will vary the protease incubation time and utilize the techniques developed to determine the resolution of this curvature-based analysis in identifying biomechanical changes resulting from a range of elastin degradation. Porcine aortic valve leaflets are structurally similar to human specimens, and we expect that the resolution limits identified in these studies would be relevant for utilization in a clinical setting. The absolute measures of curvature that may detect the onset and progression of CAVD, however, would require extensive validation for human patients using relevant imaging modalities.

## Figures and Tables

**Figure 1 bioengineering-06-00039-f001:**
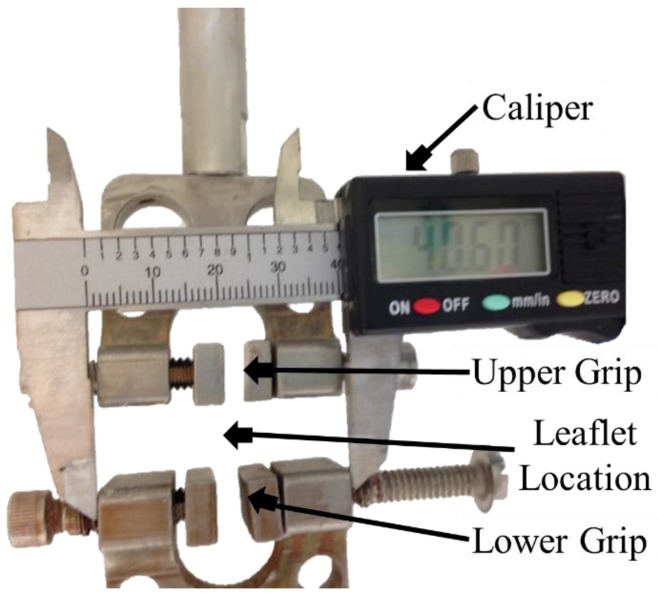
Leaflet strip set-up in mechanical test instrument (Electroforce 2300 with Wintest 7 software control, TA instruments) to facilitate specimen cyclic flexure (bending and straightening events).

**Figure 2 bioengineering-06-00039-f002:**
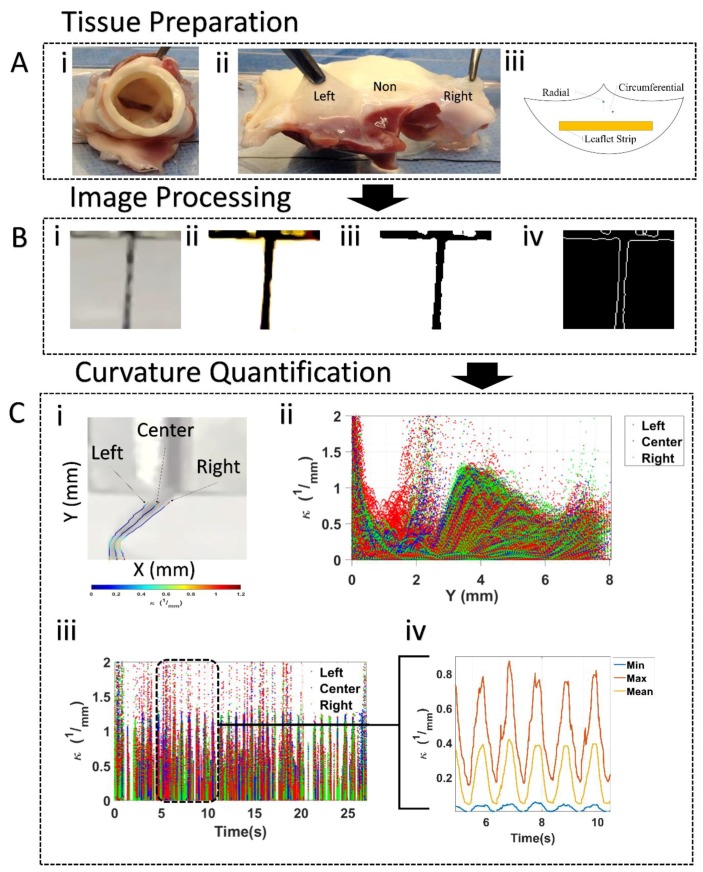
Workflow (**A**) Tissue preparation: i) Porcine aortic valve, ii) split aortic valve revealing the three leaflets, iii) schematic of circumferentially oriented rectangular strips cut from the belly region of porcine aortic valve leaflets. The remaining surrounding tissue was minced and used for the degradation and quantification of elastin in this study. (**B**) Image processing: i) Cropped image according to region of interest, ii) increase background to foreground contrast, iii) threshold image, iv) edge detection. (**C**) Curvature quantification: i) Left/center/right curvatures for a frame of a video, ii) temporal curvature for a given leaflet, iii) spatial curvature for a given leaflet, iv) minimum, Maximum, and mean curvature along center profile for the 5–10 range.

**Figure 3 bioengineering-06-00039-f003:**
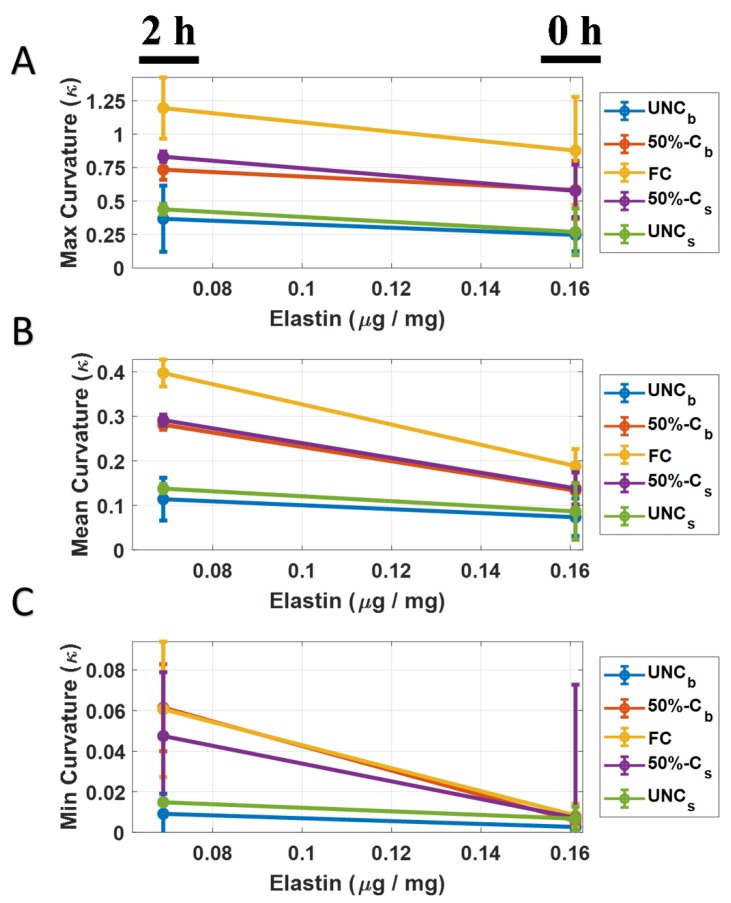
Elastin and curvature comparisons. (**A**) Maximum curvature with mean ± SEM plotted against elastin for 0 h and 2 h degradation time points for all bending phases. (**B**) Mean curvature with mean ± SEM plotted against elastin for 0 h and 2 h degradation time points for all bending phases. (**C**) Minimum curvature with mean ± SEM plotted against elastin for 0 h and 2 h degradation time points for all bending phases.

**Figure 4 bioengineering-06-00039-f004:**
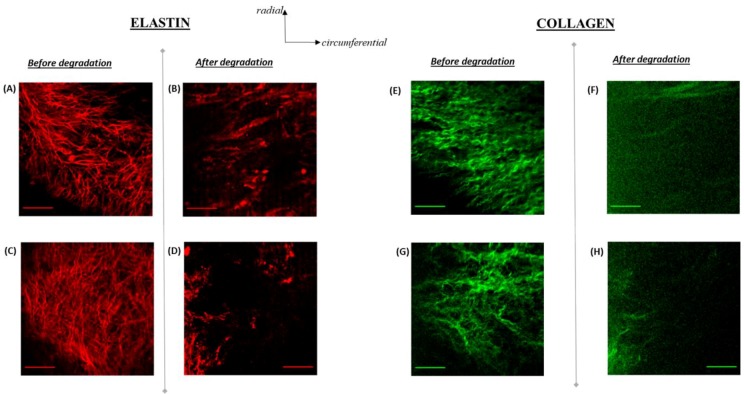
(**A**–**D**) Elastin structure in aortic valve leaflets before (**A**,**C**) and after (**B**,**D**) elastase-degradation. Images (**A**) and (**B**) were obtained from the non-coronary cusp whereas images (**C**) and (**D**) were from the left coronary cusp. Image sections were taken at a depth of 36 µm from the nearest leaflet surface. Scale bar in all images is 50 µm. Elastin fibers (**A**,**C**) exhibited a strong affinity for alignment in the radial direction. Enzymatic-degradation clearly led to loss of elastin content, orientation and network in the leaflets (**B**,**D**). (**E**–**H**) Collagen distribution at the same spatial location in which Elastin structure was imaged (**A**–**D**). Collagen fibers (**E**,**G**) were primarily oriented in circumferential direction. Degradation with elastase enzyme was not specific to elastin as clearly, loss of collagen content, orientation and network in the leaflets also occurred (**F**,**H**).

**Figure 5 bioengineering-06-00039-f005:**
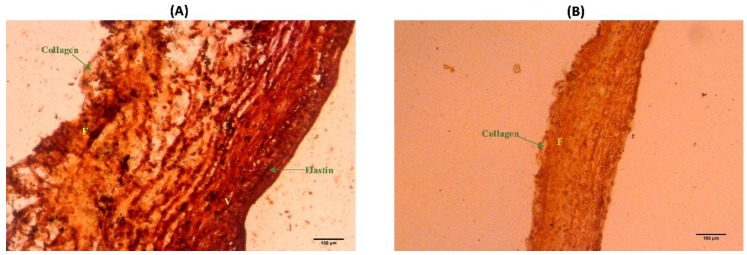
Movat’s staining of: (**A**) Normal native porcine aortic valve non-coronary leaflet and (**B**) elastase-degraded (0.5 mg/mL for 2 h) native porcine aortic valve non-coronary leaflet. The extracellular matrix (ECM) components stain as brownish-black for Elastin and yellow-orange for Collagen. “F” indicates the fibrosa side and “V” is the ventricularis side of the leaflet. Note that the ventricularis side of the leaflet was completely lost after elastase degradation (B).

**Table 1 bioengineering-06-00039-t001:** Bending/straightening behavior of samples with and without 2 hours (2 h) of elastin degradation.

Group	UNC_b_	50%-C_b_	FC	50%-C_s_	UNC_s_
**Control 0 h**	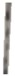	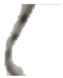	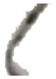	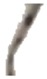	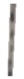
**2 h**	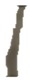	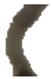	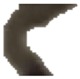	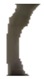	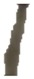

**Table 2 bioengineering-06-00039-t002:** Mean ± standard error of the mean (SEM) of bending of samples of mean curvature, k_mean_ (1/mm) at five levels of bending for the un-degraded and elastin-degraded porcine aortic valve leaflet specimens after 2 h of elastase exposure (n = 6–9 samples/group). UNC_b_, 50%-C_b_, stands for unconstrained and semi-constrained during leaflet strip bending, while UNC_s_ and 50%-C_s_ refer to these events during specimen straightening; FC stands for the leaflet strip at the fully constrained or fully-flexed state.

Group	Amount of Bending					n
Elastin Degradation	UNC_b_	50%-C_b_ *	FC *	50%-C_s_ *	UNC_s_	
**0 h (Control)**	0.07 ± 0.04	0.13 ± 0.04	0.19 ± 0.04	0.14 ± 0.04	0.09 ± 0.06	9
**2 h**	0.11 ± 0.05	0.28 ± 0.01	0.40 ± 0.03	0.29 ± 0.01	0.14 ± 0.01	6
***p*-Value**	*p* > 0.05	*p* < 0.05	*p* < 0.05	*p* < 0.05	*p* > 0.05	

* Indicates significance when compared to its corresponding control value.

**Table 3 bioengineering-06-00039-t003:** Mean ± SEM of bending of samples of min curvature, k_min_ (1/mm) at five levels of bending for the un-degraded and elastin-degraded porcine aortic valve leaflet specimens after 2 h of elastase exposure (n = 6–9 samples/group). UNC_b_, 50%-C_b_, stands for unconstrained and semi-constrained during leaflet strip bending, while UNC_s_ and 50%-C_s_ refer to these events during specimen straightening; FC stands for the leaflet strips at the fully constrained or fully-flexed state.

Group	Amount of Bending					n
Elastin Degradation	UNC_b_	50%-C_b_ *	FC	50%-C_s_	UNC_s_	
**0 h(Control)**	0.0026 ± 0.0006	0.0055 ± 0.0017	0.0083 ± 0.0058	0.0071 ± 0.0656	0.0067 ± 0.0059	9
**2 h**	0.0091 ± 0.0099	0.0614 ± 0.0214	0.0606 ± 0.0334	0.0475 ± 0.0313	0.0148 ± 0.0001	6
***p*-Value**	*p* > 0.05	*p* < 0.05	*p* >0.05	*p* >0.05	*p* > 0.05	

* Indicates significance when compared to its corresponding control value.

**Table 4 bioengineering-06-00039-t004:** Mean ± SEM of bending of samples of max curvature, k_max_ (1/mm) at five levels of bending for the un-degraded and elastin-degraded porcine aortic valve leaflet specimens after 2 h of elastase exposure (n = 6–9 samples/group). UNC_b_, 50%-C_b_, stands for unconstrained and semi-constrained during leaflet strip bending, while UNC_s_ and 50%-C_s_ refer to these events during specimen straightening; FC stands for the leaflet strips at the fully constrained or fully-flexed state.

Group	Amount of Bending					n
Elastin Degradation	UNC_b_	50%-C_b_	FC	50%-UC_s_	UNC_s_	
**0 h (Control)**	0.25 ± 0.12	0.58 ± 0.22	0.88 ± 0.40	0.58 ± 0.20	0.27 ± 0.17	9
**2 h**	0.37 ± 0.25	0.73 ± 0.08	1.20 ± 0.23	0.83 ± 0.04	0.44 ± 0.04	6
***p*-Value**	*p* > 0.05	*p* > 0.05	*p* > 0.05	*p* > 0.05	*p* > 0.05	

**Table 5 bioengineering-06-00039-t005:** Mean ± SEM elastin concentration in un-degraded and elastin-degraded porcine aortic valve leaflet specimens after 2 h of elastase exposure. (n = 3 samples/group).

Group	Elastin Degradation	
Elastin (µg/mg)	**0 h (Control)**	**2 h**
	0.161 ± 0.07	0.069 ± 0.016

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
