# Peer review of "Elastin-Dependent Aortic Heart Valve Leaflet Curvature Changes During Cyclic Flexure"

_bioengineering, 2019, doi:10.3390/bioengineering6020039_

Round 1
Reviewer 1 Report
In this manuscript, the authors have examined how alterations in aortic valve leaflet curvature are associated with degradation of the valve’s elastin-rich ECM using porcine aortic valves that undergo degradation by elastase, followed by cyclic flexural mechanical testing. In general, this is a promising and interesting study that requires several important clarifications and additional data before it is suitable for publication.
-The authors motivate their work with the notion that mean curvature measurement in humans could be utilized as a clinical biomarker of early elastin degradation. Do current clinical imaging techniques have sufficient spatial/temporal resolution for such mean curvature measurements? Presumably, mean leaflet curvature will be impacted by factors with high patient-to-patient variability, such as blood pressure and aortic root configuration/anatomy. The authors should comment on how these aspects of the in vivo environment would be taken into account.
-How many donors were used in cyclic flexure experiments? Do the 9 and 6 strips per condition imply 9 donors, or 3 donors x 3 valve leaflets? There is notable leaflet-to-leaflet (and donor-to-donor) heterogeneity in a number of cellular and biomechanical aspects of aortic valve biology – did the authors note, measure, or account for such differences between the left, right, and non-coronary leaflets in terms of curvature?
-Regarding the usage of enzymatically-induced elastin degradation: Is the degree of degradation physiological? Though the low resolution makes it difficult to be certain, Figure 4 indicates that the elastase treatment does not just fragment elastin (as is the case in CAVD), but also dramatically reduces total elastin content. The authors state that “Therefore, it is important to note that an increase in the axial 1-D mean leaflet curvature measurement in elastin-degraded leaflets is due to some combination of losses in both bulk elastin content and its structure”, but Fig4 would seem to imply that this signal is largely (almost exclusively?) due to elastin loss. Namely, is it consistent with that found in early (or late) CAVD remodeling? If not, then the relevancy of significant differences in mean curvature between control and enzymatically degraded leaflets to human CAVD must be clarified. Quantification of IHC elastin staining and degree of fragmentation in human CAVD samples and porcine degraded samples would answer this. In the same vein, are the dimensions, mechanical properties, and elastin content of the ventricularis similar between porcine and human valves?
-In Table 1, the 2-hour (elastase) group leaflet appears notably thicker than the control leaflet. Were leaflet dimensions recorded before degradation, after degradation, and before/after cyclic testing? If so, were there significant differences in leaflet thickness at any point between control and elastase-treated leaflets? Such differences in morphology could impact curvature and flexure. Was isosmotic solution used during the 2-hour elastin degradation? Did degradation occur immediately prior to cyclic testing? Did the 0 hr control group undergo a mock degradation for 2 hours in elastase-free PBS? Please clarify and account for what has occurred.
Author Response
Kindly refer to attached file. Thanks.

Reviewer 2 Report
The manuscript entitled “Elastin-dependent leaflet curvature changes during cyclic flexure: relevance to aortic valve calcification” reports on an interesting correlation between the amount of tissue bending and the elastin content. In specific, the results indicate that the bending curvature increases with elastin degradation. This is an interesting observation; however, I’m left but wondering whether correlation necessarily means causation.
The manuscript is nicely presented and I think will make a positive contribution to the literature. I would only make the following comments for consideration by the authors in their revision.
Major concerns
1- The study shows that reduction is elastin content will increase the bending curvature. However I would urge caution in drawing the reverse conclusion, in that the increased amount of flexion (bending) does not necessarily mean that the elastin content in the leaflet has been reduced. It may well be the case that collagen degradation has happened instead. From a (bio)mechanics perspective, many structural defects can result in increase in bending, or decrease in bending rigidity. I suggest that the authors tone-down their narrative in this regard, and bring to the attention of the reader that the increase in bending curvature may not be an accurate measure of the elastin content/degradation in the tissue, as degradation of/damage to other structural constituents of the valve such as the collagen architecture can also result in the increase in bending.
2- There is a body of literature addressing the effects of rate of deformation on rigidity of the valvualr tissue; in effect showing that if the tissue is deformed faster it will intrinsically show a stiffer behaviour. These studies include:
Bell et al. (2018), J. Biomech. Eng., doi: 10.1115/1.4039625, for arterial tissues.
Karunaratne et al. (2018), Scientific Reports, doi: 10.1038/s41598-018-21786-z, for ligaments.
Anssari-Benam et al. (2017), Royal Society Open Science, https://doi.org/10.1098/rsos.160585.
Anssari-Benam et al. (2019), Acta Biomaterialia, https://doi.org/10.1016/j.actbio.2019.02.008.
Therefore, even in an elastin-degraded cusp, if the tissue is loaded faster than 1Hz frequency, a reduced amount of bending/flexion rigidity will be observed. This again indicates that the amount of bending rigidity/curvature may not be a direct result of only the reduction in elastin or any other structural constituent. Instead, it is also influenced by the loading boundary condition. I suggest that the authors acknowledge the studies showing rate effects in valves, such as above, and confer the conclusion that loading boundary conditions such as rate effects will also influence the observed bending curvature and rigidity.
Minor observations
1- In the title of the article, the letter ‘d’ in ‘during’ should be in capital: ‘During’.
2- Was the sample number in this study 3? The authors appear to indicate that the number of samples used in this study is only – Page 2, Line 73: “(n = 3 valves)”.
3- Y_prime and Y_double prime in equation (2) should in fact be f_prime and f_double prime, to be consistent with the notation used in equation (1), and also to reflect the variable with respect of which Y should be differentiated.
Reviewer 3 Report
The study of Tesfamariam et al., focuses on the effect of elastin disruption on the curvature changes of the aortic valve. Slices of porcine aortic leaflets are exposed to elastase and subjected to cyclic flexure. They found that loss of elastin integrity resulted in an increase of curvature, which led to the suggestion that curvature measurements can serve as a potential biomarker for elastin degradation.
It is known that elastin disruption is an important feature of aortic valve disease, which might lead to further pathological remodelling of the valve and therefore to changes of mechanical properties of the aortic valve. The results of the present study fits with this idea.
There are however a few elements that would require clarification.
The images in table 1 show that the elastase treated group has greatly thickened leaflets. No mentioning or explanation is given in the text.
Considering the interplay between collagen and elastin, the loss of elastin might lead to disruption of collagen organization and maybe of other components in the leaflet (as can maybe be suggesting from the thickening of the leaflets). The resulting changes in curvature is therefore likely the consequence of the complete set of changes happening in the leaflet and not merely of elastin disruption. Has the collagen and other ECM components been examined to clarify this?
Related to this, curvature measurements as a biomarker for elastin degradation has been suggested. However, more than just elastin degradation is present in the tested leaflets (see above). Further, the extent of elastin degradation is so much that changes in curvature can be observed. In the onset of CAVD the elastin degradation is much less and local that changes are probably not found until the disease is in an advanced stage. So more experiments with less elastin disruption need to be tested to support that suggestion.
Further, although mentioned in limitations, because of the use of circumferentially dissected strips the deformations do not reflect the in vivo situation. Do the authors have indications/ results that show that their findings are similar as they would have used the radially dissected strips?
The mentioning of “relevance to aortic valve calcification” in the title is too premature based on the findings.
Has the control group been incubated in PBS for 2 hrs at 37 degrees Celsius similar to the treated group, but then without elastase?
Round 2
Reviewer 1 Report
The authors have made important clarifications to their methodological approaches which directly and clearly address the limitations of their elastase-induced degradation. As the study stands, and as the authors admit, it remains difficult to dissect the relative contributions of elastase degradation, collagen degradation, and leaflet thickening to alterations of bending curvature. Without completely re-performing the experiments in this manuscript from scratch, determining the relative contributions of these factors will be very challenging. Regardless, the authors have identified an important biomechanical phenomenon that occurred in response to proteolytic valve leaflet degradation, and have done a suitable job of describing the limitations of their approach. Perhaps future studies will incorporate additional controls that enable further understanding of how these three factors impact leaflet curvature.
Author Response
We thank you for your very useful comments and suggestions.
Reviewer 3 Report
The authors made some clear changes in the text especially regarding the limitations. What is troublesome is the effect of the elastase. Elastin content is down, however, also the collagen content. Collagen integrity depends on elastin, but after 2 hours should still be there, although maybe misaligned. If not or much less, as appears from figure 4, the question rises, what are we looking at after the elastase treatment? This is an important question as the cyclic flexure experiments are performed on this tissue. Did the authors make sections through the treated and untreated samples after the flexure experiments with an orientation the traditional 3 layers can be distinguished? Were they stained for elastin, collagen, GAGs? Movat pentachrome staining or Weigerts Resorcin Fuchsin, Masson's Trichrome, Alcian Blue, antibody stainings might help. Then maybe conclusions may be drawn whether what made curvature changes happen.
Author Response
Kindly see attached document. Thank You.
